# Nondestructive Determination of Epicarp Hardness of Passion Fruit Using Near-Infrared Spectroscopy during Storage

**DOI:** 10.3390/foods13050783

**Published:** 2024-03-03

**Authors:** Junyi Wang, Dandan Fu, Zhigang Hu, Yan Chen, Bin Li

**Affiliations:** 1College of Mechanical Engineering, Wuhan Polytechnic University, Wuhan 430023, China; wangjunyidyx@163.com (J.W.);; 2College of Food Science and Technology, Huazhong Agricultural University, Wuhan 430070, China

**Keywords:** passion fruit epicarp, near-infrared spectroscopy, hardness, Grids-RF model, GA-SVR model

## Abstract

The hardness of passion fruit is a critical feature to consider when determining maturity during post-harvest storage. The capacity of near-infrared diffuse reflectance spectroscopy (NIRS) for non-destructive detection of outer and inner hardness of passion fruit epicarp was investigated in this work. The passion fruits’ spectra were obtained using a near-infrared spectrometer with a wavelength range of 10,000–4000 cm^−1^. The hardness of passion fruit’s outer epicarp (F1) and inner epicarp (F2) was then measured using a texture analyzer. Moving average (MA) and mean-centering (MC) techniques were used to preprocess the collected spectral data. Competitive adaptive reweighted sampling (CARS), successive projection algorithm (SPA), and uninformative variable elimination (UVE) were used to pick feature wavelengths. Grid-search-optimized random forest (Grids-RF) models and genetic-algorithm-optimized support vector regression (GA-SVR) models were created as part of the modeling process. After MC preprocessing and CARS selection, MC-CARS-Grids-RF model with 7 feature wavelengths had the greatest prediction ability for F1. The mean square error of prediction set (RMSE_P_) was 0.166 gN. Similarly, following MA preprocessing, the MA-Grids-RF model displayed the greatest predictive performance for F2, with an RMSE_P_ of 0.101 gN. When compared to models produced using the original spectra, the R^2^_P_ for models formed after preprocessing and wavelength selection improved. The findings showed that near-infrared spectroscopy may predict the hardness of passion fruit epicarp, which can be used to identify quality during post-harvest storage.

## 1. Introduction

Passion fruit belongs to the category of leapfrog fruits, exhibiting rapid post-harvest changes and manifesting characteristics of short shelf life, quick wrinkling, and post-ripening [1,2]. Research on the quality changes of passion fruit during storage primarily focuses on comparing the effectiveness of various packaging technologies. Parameters measured include respiration intensity, size, mass, color, and internal chemical composition. The assessment of peel changes relies heavily on colorimetric analysis and the proportion of shrinkage area observed through manual observation [3,4]. The passion fruit peel, serving as an external protective layer, plays a crucial role in the ripening and storage freshness of passion fruit. The industry explores the secondary utilization of passion fruit peels; however, non-destructive testing research on peel hardness changes during storage has not been initiated. As a byproduct for extraction materials, passion fruit peels hold tremendous potential, necessitating in-depth investigation into their quality attributes.

Phenomena related with the processes of development, atrophy, and aging of biological tissues are linked to complicated biochemical and physical mechanisms that lead to changes in textural characteristics [5]. In pursuit of attaining non-destructive quality assessment of biological tissues, a diverse array of detection methods is extensively employed, comprising optical, acoustic, and other techniques. Hardness stands out as a crucial factor for evaluating the quality of fruit peel [6]. The usefulness of assessing fruit or peel hardness as the main index for judging ripeness has been extensively accepted in numerous processing businesses. For example, hardness is an important quality factor applied in the apple processing sector to separate fruit batches. Hardness, comprising crispness, offers consumers adequate textural features and acts as an assessment criterion for ripeness and storage capability throughout the post-harvest phases [7]. 

Expanding on the concept of hardness as a quality indicator, it plays a significant role when assessing the quality of stored fruit. The hardness of a fruit, related to its composition, serves as a reliable indicator of maturity [8]. When complemented with other quality metrics, fruit hardness can effectively aid in the post-harvest quality evaluation. Furthermore, the hardness of the passion fruit skin has been identified as one of the key determinants of the fruit’s freshness [9]. From quality assessment to management, understanding and regulating hardness changes in passion fruit during storage can prove valuable. Developing a model to perpetually monitor the quality of passion fruit could potentially pave the way for improved storage conditions and consequently, an extended shelf-life. When considering logistics and waste management, hardness measurement for passion fruit can be effectively utilized as a measure to confirm the fruit’s mechanical resistance during transit, which could potentially minimize waste during processing.

Traditional hardness detection techniques usually depend on the subjective manual labor of expert individuals. The employment of near-infrared spectroscopy technology in hardness detection has grown more essential. This technology quantitatively examines hardness by detecting information about material composition and internal molecular structure, enabling non-destructive evaluation of fruit hardness. The hardness of fruits is controlled by characteristics such as density, cell structure, and the degree of pectin breakdown [10]. The near-infrared spectral region comprises information regarding molecular structure and compositional condition, assisting to the analysis of the physicochemical quality of materials. Near-infrared spectroscopy technology permits for a more accurate and objective assessment, notably in critical physicochemical parameters such as fruit texture hardness, maturity, and nutritional components. 

Near-infrared spectroscopy (NIRS) has grown into an important tool for assessing fruit firmness. Fei Tan et al. employed NIRS to examine tomato hardness in order to determine ripeness. To determine essential quality characteristics of cherry tomatoes, they employed a multispectral imaging system in conjunction with visible/near-infrared spectroscopy and near-infrared spectroscopy [11]. Jason Sun et al. separately investigated the possibilities of a NIRS system and a multispectral imaging system for high-speed hardness grading of apples. They evaluated two spectroscopic systems, and the findings suggested a somewhat improved measuring performance for NIRS system, with the best correlation [12]. Fengjiao Ping et al. have effectively constructed excellent spectrum models by investigating grape hardness and other physicochemical properties, thus providing a valuable technique for quick, non-destructive detection in measuring grape maturity, enhancing quality control [13]. These case studies further support the viability and accuracy of NIRS technology in measuring fruit hardness.

Currently, minimal work into the non-destructive detection of hardness changes in the epicarp during storage utilizing NIRS exists. This research intends to investigate the non-destructive detecting capabilities of near-infrared diffuse reflectance spectroscopy for epicarp hardness of passion fruit during storage. The objectives of this study were as follows: (1) to analyze the variation patterns of epicarp hardness on both sides of passion fruit during different storage periods; (2) to select the optimal feature wavelength combinations that characterize the hardness of the epicarp on both sides; (3) to establish the optimal prediction model for epicarp hardness based on near-infrared spectroscopy. 

## 2. Materials and Methods

### 2.1. Experimental Sample

For this experiment, 120 passion fruits were selected from an Ali passion fruit digital agricultural situated in the city of Kunming, Yunnan Province, China. Freshly picked from multiple farms in the province, these passion fruits had a deep purple skin color and weighed between 54 ± 6 g. There was no visible evidence of insect or mechanical damage. After cleaning the dust and contaminants from the surface of passion fruits, the gathered fruits were blended and stored in a controlled environment at a constant temperature of 25 °C and 85% relative humidity. To conduct a random sampling experiment, a batch of 20 passion fruits was picked at random every three days, for a total of six batches during the course of the trial.

### 2.2. Near-Infrared Spectra Acquisition

The samples were subjected to diffuse reflectance spectral data collection utilizing a near-infrared spectral (NIRS) acquisition instrument (Frontier NIR with Integrating sphere measurements, PerkinElmer Spectrum Two company). Figure 1 depicts the spectrum acquisition system, which is separated into two pieces, with the aluminum plate serving as a smaller collecting window to guarantee that only the near-infrared light traveling through the hole is captured. On this premise, the sample holder reduces the effect of out-of-hole information and helps to retain the sample in place. In the test, a single passion fruit is positioned on a sample platform, ensuring that the point to be measured aligns with the beam. Measurements are conducted within a light-isolated black box to eliminate external light interference. The NIRS were acquired in the range of 10,000–4000 cm^−1^ (1000–2500 nm) with a resolution of 4 cm^−1^ and 8 repeat scans. To thoroughly identify the spectrum information of the whole passion fruit, three sample points are consistently picked along the equatorial plane of each sample, yielding absorbance spectral curves corresponding to these three spots. The average of the three spectral curves is used as the spectral data for the given sample, and the following analysis is undertaken.

### 2.3. Texture Measurement

The penetration test is a commonly used physical property detection technique, typically utilized for objects that may vary in shape and size but possess reproducible features. Accurate uniform measurements for these types of objects are often difficult to accomplish, and the penetration test provides a solution to this problem. By measuring penetration force, we can objectively evaluate the characteristics of different objects’ internal structures and textures [14]. In the field of food science, penetration testing has been successfully applied to the hardness studies of various fruits, including tomatoes, grapefruits, and mangoes [15,16,17]. Generally, evaluating the maximum peak (maximum load-bearing capacity) of the force-displacement curve is a common way to represent fruit hardness [18]. This indicator reflects the fruit’s resistance to external force and is usually used to represent fruit hardness. The penetration test method and hardness calculation mentioned above were referred to in this study. According to the actual characteristics of passionfruit, the specific parameters were set.

The samples were subjected to puncture studies using a texture analyzer (TA.XTC-18 Texture Analyzer, Shanghai Baosheng Industrial Development Co., Ltd., Shanghai, China) equipped with a needle probe (TA/2N needle probe), as illustrated in Figure 2. To fully capture the characteristics of passion fruit, six equidistant locations were selected along the equatorial line of each fruit for puncture testing. This was done due to the complicated surface and triangular interior distribution of passion fruit. The specimens were positioned onto the experimental platform and subsequently penetrated using a stainless-steel probe measuring 1 mm in diameter. The parameters for the texture analyzer were configured as follows: an inductive force of 0.05 N, a puncture speed of 0.1 mm/s, and a puncture distance of 10 mm. Force–displacement curves were plotted for each sampling point of the individual samples. The first and second peak forces during the puncture process of the probe penetrating the sample on each curve were calculated, and force was measured in gN. The calculated values were averaged to obtain the characteristic values for individual samples. The subsequent analysis was based on the average of six measurements for each individual sample.

### 2.4. Multivariate Data Analysis

#### 2.4.1. NIRS Sample Set Partitioning

Sample partitioning was conducted using sample set partitioning based on Joint X-Y Distance (SPXY) Algorithm. Building upon the Kennard–Stone algorithm, the SPXY algorithm incorporates observed values as the basis for partitioning, reducing the impact of irrelevant information from spectral data on the results, minimizing random errors and enhancing the stability of the model.

#### 2.4.2. NIRS Data Preprocessing

The spectrum-smoothing procedure can efficiently remove random mistakes in spectral data, resulting in better spectral data stability. The moving average method (*MA*) is a popular smoothing technique. As indicated in Equation (1), the concept is to replace the original values at the center wavelength point with the average value derived from the specified window. The smoothing process is implemented across all positions by moving the window.
(1)$$X_{MA\;}=\frac{1}{2w+1}\sum_{i=-w}^{+w}X_{k+i}$$
where $X_{MA\;}$ represents the smoothed spectral values, *k* is the position of the center wavelength point, and *W* is the size of window.

The notion of mean centering (MC) refers to subtracting the average spectrum of a calibration set from the spectra of individual samples. The application of MC improves the differences across sample spectra and links spectral variations with changes in analytical substance, hence emphasizing distinguishing characteristics of the variation. This approach increases the performance of the built model.

#### 2.4.3. Effective Wavelength Selection

Effective Wavelength Selection, in essence, involves feature selection for spectral data. Spectral data often contains a considerable amount of redundant information and noise, which can negatively impact the performance of machine learning algorithms. Appropriate sample selection serves to reduce model complexity and training time, thereby mitigating the risk of overfitting.

The Competitive Adaptive Reweighted Sampling (CARS) Algorithm employs the partial least squares regression (PLS) method to calculate importance weights for each feature and subsequently normalizes these weights. Features with higher standardized weights are selected, while those with lower weights are excluded. The calculation of weights is defined by Formula (2). An adaptive approach is then employed to select the most crucial features. This process is iteratively repeated until reaching the utmost number of iterations or attaining the accuracy criteria defined via cross-validation.
(2)$$w_{i}=\frac{\left|b_{i}\right|}{\sum_{n=1}^{p}|b_{i}|}$$
where *W_i_* is the weight of the *i*th vector and *b_i_* is the *i*th vector of regression coefficients for PLS.

The successive projection method (SPA) also utilizes PLS to select relevant characteristics in the data. It achieves this by developing the feature set gradually until the requisite precision is reached. For an originally picked wavelength variable, its influence on the other remaining unselected wavelength variables in the subspace is calculated from the projected values. The wavelength variable that matches the maximum value in the projected value is then added to the chosen features. This procedure is performed until the required number of features is acquired. The formula for computing the projection values is provided below:(3)$$p=x_{j}-(x_{j}^{T}x_{k})x_{k}{\left(x_{k}^{T}x_{k}\right)}^{-1}$$
where *p* is projection value, $x_{j}\;$is the chosen wavelength column vector, and $x_{k}\;$is wavelength column vectors in the residual subspace.

To increase its complexity, the uninformative variable elimination (UVE) technique mixes random noise data into the original spectral data. The combined dataset, which includes the extra noise data, is used to build a partial least squares model. The regression coefficient matrix is created by iterative sample elimination. Each wavelength feature’s stability is examined, and a threshold for wavelength selection is established. Wavelengths with stabilities greater than the criterion are kept as feature variables. The stability is calculated using the following formula:(4)$$h_{i}=\frac{\mathrm{mean}(b_{i})}{\mathrm{std}(b_{i})},i\in [1,2n]$$
where $h_{i}$ is the *i*th variable stability and $b_{i}$ is the ith variable regression coefficient.

#### 2.4.4. Random Forest Model of Grid Search Algorithm Optimization

Random forest (RF) is an ensemble learning methodology that leverages multiple decision trees. Tuning of key hyperparameters, such as the number of decision trees (ntree) and the number of features per node (mtry), is pivotal to balancing between overfitting and underfitting.

Enhancement of the RF model is achieved through grid search, an exhaustive technique yielding optimal parameters via all possible combinations of ntree and mtry values. Five-fold cross-validation is used to manage issues such as overfitting and sample distribution. It segments the dataset into five subsets and iteratively employs each as a test set with the others serving for training. 

The fundamental stages for augmenting the RF (random forest) model via grid search are outlined as follows: (1)Grid search is employed to explore a wide range of combination values of the parameters “ntree” and “mtry”.(2)K-fold cross-validation, a fundamental element in the RF training process, is used to calculate the average error of the k results. This average error is referred to as the cross-validation error of the RF model.(3)An iterative process is initiated across the “ntree” and “mtry” parameters, systematically applying varying combinations of parameter values for RF training and simultaneously calculating the cross-validation error for each iteration.(4)The RF model associated with the smallest cross-validation error is selected as the optimized model.

In this study, the algorithms were implemented using MATLAB Version 1.8.0 (R 2019a) software. In the Grids-RF (5-folds) model, we utilized a random forest regression model provided by Abhishek Jaiantilal’s randomforest-matlab project accessible under the GNU GPL v2 license and hosted by Google Code (https://code.google.com/archive/p/randomforest-matlab (accessed on 21 September 2023)). We applied a five-fold cross-validation method and employed a grid-search to optimize the “ntree” and “mtry” parameters.

#### 2.4.5. Support Vector Regression Model of Genetic Algorithm Optimization

The support vector regression (SVR) technique leverages nonlinear relationships in data through an analytical space defined by the spatial distance between sample points. The radial basis function, acting as the Gaussian kernel, is mathematically represented as:(5)$$K(x_{i},x_{j})={e^{–\gamma \|x_{i}–x_{j}\|}}^{2}$$

In SVR, the “c” and “γ” parameters are crucial. “c” balances model simplicity and data fit, while “γ” adjusts the influence of each sample on the model to refine decision boundaries.” To optimize these parameters, the genetic algorithm (GA), drawing on principles of biological evolution, iteratively seeks optimal “c” and “γ” values, enhancing the efficiency of the SVR model. The underlying process for enhancing the SVR model by employing the parameter optimization through the GA is as follows:(1)An initial population is constructed, which encompasses a diverse array of parameter combinations for “c” and “γ”.(2)The pre-configured settings are applied to each corresponding set of “c” and “γ” for training the SVR model. The mean squared error (*MSE*) is formulated as the fitness measure through the calculation:
(6)MSE=∑in(y^i−yi)2n
where${\;y}_{i}\;$is observed value of the sample, y^i is predicted value, and *n* is sample size.

(3)Samples of high fitness are selected, advantageous characteristics from multiple samples are integrated using crossover, and unpredictability is injected by performing mutation operations, preventing the model from falling into local optima. The population size is kept consistent by replacing lower-fitness samples with new ones.(4)The optimization process—including the preservation of a consistent population size, the crossover and mutation operations, as well as selection mechanisms—is repeated until the fitness measure stabilizes and no longer decreases.(5)This iterative process results in the emergence of “c” and “γ” values associated with the minimal *MSE*.(6)The final optimized SVR model is then obtained by using the “c” and “γ” values determined in Step 5.

In this study, the algorithms were implemented using Matlab 2019a software. The GA-SVR model was established based on the LIBSVM toolbox, which was developed by Professor Chih-Jen Lin and his team at National Taiwan University. We employed the Radial Basis Function (RBF) as the kernel for SVR and used a genetic algorithm to determine the optimal penalty factor “c” and kernel function parameter “γ”.

#### 2.4.6. Model Evaluation Indicators

Performance assessment metrics were computed based on the outcomes of computing the corrective set and prediction set. Correlation coefficient of calibration/prediction (*R*^2^*_C_*/*R*^2^*_P_*), root mean square error of calibration/prediction set (*RMSE_C_*/*RMSE_P_*), and residual predictive deviation (*RPD*) are selected for evaluation indicators. The formulas are as follows [19]:(7)R2C(R2P)=∑i=1n(yi−y^)2/∑i=1n(yi−y¯)2
where $y_{i}\;$ is the standardized value of the observations in the *i*th sample, y ^ is the projected value of the ith sample, and $\overset{¯}{y\;}$ is the mean of the sample observations.
(8)RMSEC(RMSEP)=1n∑i=1n(yi−y^i)2

The ratio of performance to deviation, or *RPD* value, was used to assess all calibrations. When the *RPD* is between 2 and 2.5, it suggests that the model can be roughly quantitatively assessed, and when *RPD* is larger than 2.5, it shows that the model has a strong predictive impact.
(9)$$RPD=\frac{SD}{{RMSE}_{P}}$$
where *SD* is the standard deviation of the prediction set.

## 3. Results

### 3.1. Analysis of Patterns of Variation in Textural Parameters

The pericarp of the passion fruit can be classified into three distinct layers, namely the exocarp, mesocarp, and endocarp. The external surface of the exocarp, also known as the outer epicarp (P1), is characterized by its smooth and leathery texture. The inner epicarp (P2), which is closely adhered to it, can be identified as a deep purple layer. This layer corresponds to the internal component of the exocarp, namely the region inside the epicarp. Additionally, the mesocarp, characterized by its white and spongy texture, extends beyond the deep purple layer. The endocarp, a gelatinous and semi-transparent layer that is internally yellow, surrounds the black seeds of the passion fruit. This layer acts as the primary consumable component of the fruit [20].

Figure 3 shows the texture characteristic curves for different storage times. The first peak in the curve (NO. 1 peak,) reflects the physical properties when the probe penetrated the P1. At this point, the pericarp tissue undergoes instantaneous fracture, and the observed peak is the maximal puncture force (F1). F1 can be characterized as the hardness of the outermost layer of the fruit pericarp.

The second peak (NO. 2 Peak) reflects the physical properties when the probe penetrated the deep purple layer, which constitutes P2. At this point, the observed peak is the maximal puncture force (F2).

After the NO. 2 Peak, the curve shows a stable phase. This is due to the fact that the mesocarp, which constitutes the middle layer of the pericarp, is a porous spongy tissue, resulting in a phase where there is no immediate penetration of the tissue and no distinct peak.

The puncture forces, F1 and F2, corresponding to the two peaks in the texture analysis of each sample were calculated. Table 1 shows the results of a descriptive statistical analysis performed on F1 and F2. F1 and F2 show quite substantial standard deviations. F1 has the highest standard deviation of 0.574, while F2 has the lowest standard deviation of 0.356. To some degree, the large variability suggests major changes throughout the storage process, which is important for study. This is mostly due to changes in the tissue structure of the passion fruit during storage.

As shown in Figure 4, the texture profile analysis demonstrates regular fluctuations in puncture forces, F1 and F2, which have a negative connection with storage duration from days 1 to 13. F1 shows a significant increase from days 13 to 16, but F2 continues the pattern seen in the first 13 days. F2 fluctuates significantly between days 1 and 13, with the average value of F2 for the sample on the first day of storage being 4.109 gN and dropping to 2.537 gN on the 13th day. The change in F1 is rather minimal during the same storage time, with the average value for the sample on the first day of storage being 1.53 gN and reducing to 1.235 gN by the 13th day.

### 3.2. Result of Set Partitioning

The sample set is partitioned into training and testing sets using the SPXY method in a 3:1 ratio, as demonstrated in Table 2. The training set consists of 90 samples, whereas the testing set contains 30 samples. The numerical ranges of F1 and F2 in the training set are included within the range of the testing set. The mean values for F1 in the training and testing sets are 1.548 and 1.646, respectively, while the mean values for F2 in the training and testing sets are 3.432 and 4.058, respectively. This indicates that the division of the training set is appropriate.

### 3.3. Results of Spectra Preprocessing

The moving average technique (MA) and mean centering (MC) preprocessing were applied to the original spectral data, and the resultant absorbance spectra are displayed in Figure 5. Figure 5a depicts the average raw reflectance spectra of 120 passion fruits in the results and discussion section. It is evident that the spectra demonstrate a constant pattern, with substantially similar curve shapes. Significant discrepancies are seen at the absorption peaks at wavenumbers 8403 cm^−1^, 6896 cm^−1^ and 5154 cm^−1^. The spectral data suggest the potential to sensitively monitor changes in passion fruit over the storage period.

MA processed pictures of the average raw reflectance spectra of 120 passion fruits are shown in Figure 5b. The MA processing enhances the contrast between sample spectra, creating a link between spectral changes and analyte fluctuations, emphasizing the relevance of the changes. The figure shows noticeable changes in the curves, with less overlap of peak values and a greater focus on peak values among the spectral curves of various samples.

Table 3 presents a comparison of the modeling performance for the spectra after two preprocessing procedures and the original spectra. It is demonstrated that the Grids-RF model developed following MC preprocessing performs best for the F1 parameter. The R^2^_P_ and RPD values for the predictive set reach 0.898 and 2.867, respectively, indicating a considerable increase over the modeling efficacy on the original data predictive set. In terms of the F2 parameter, the Grids-RF model developed after MA preprocessing outperforms the others. After MA preprocessing, the prediction set R^2^_P_ and RPD values rise from 0.903 and 2.355 for the original spectra to 0.908 and 2.379, respectively, suggesting a significant improvement in both model accuracy and dependability. Less effective models are not considered further in the later development of spectral models.

### 3.4. Discussion of Effective Wavelength Selection

The selection of characteristic wavelengths for spectral data is based on results of a comparison of preprocessing studies and the selection of the most effective preprocessing procedures. Subsequently, the prediction performance of the GA-SVR model and the Grids-RF model based on the characteristic wavelengths obtained through different methods is studied. The wavelength results achieved using competitive adaptive re-weighted sampling (CARS), successive projection algorithm (SPA) and Uninformative Variable Elimination (UVE) are presented in Figure 6. Comparing the wavelengths selected for F1 and F2, they exhibit similar distributions around the absorption maxima at 8403 cm^−1^, 6896 cm^−1^, and 5154 cm^−1^. CARS picks the fewest wavelengths out of the three approaches, with seven and six wavelength points picked for the F1 and F2 datasets, respectively. The UVE identifies a significantly higher number of wavelengths, covering virtually all of the absorption peak sites, with 19 and 14 wavelength points picked for the F1 and F2 datasets, respectively. SPA chose 11 and 9 wavelength points for the F1 and F2 datasets, respectively, situated in the 8403–8000 cm^−1^ absorption range.

Passion fruit is a leapfrog fruit, and its internal biological processes (e.g., respiration and transpiration) result in water loss during storage. Throughout the preservation process, the passion fruit pericarp endures substantial dehydration, with water loss occurring predominantly in the pericarp rather than in the pulp [13]. Plant cell break caused by this phenomenon is a key factor contributing to the variations in hardness during storage. Therefore, the presence of spectral absorption peaks may have characterized the water loss from passion fruit epidermis.

Figure 6 shows that the absorption of water is the most important component in the spectral absorption of fruits. The strong absorption band around 5154 cm^−1^ is principally associated with the second overtone of O-H stretching, while the strong absorption band at 8403 cm^−1^ is associated with the combination of O-H stretching and deformation in water. The strong absorption band at 6896 cm^−1^ is connected with the first overtone of O-H deformation. The significant absorption of water molecules in the near-infrared region is related to the fluctuations in the curves corresponding to the development of these bands [21]. Wave numbers of roughly 8403, 6896, and 5154 cm^−1^ were screened in all of them, showing the rationality of the wave band screening.

The structure and chemical makeup of the cell wall mostly control fruit firmness. During storage, cellulose and pectin become more prone to dissolve, resulting in cell wall collapse and loosening. Changes in these components lead to variations in fruit firmness. The peel of passion fruit contains large amounts of pectin, which has garnered considerable attention due to industrial fruit processing [22]. Pectin, a fruit firmness factor, has chemical linkages such as O-H and C-H, allowing it to absorb near-infrared light. As a result, measuring the hardness of passion fruit using near-infrared reflectance spectroscopy (NIRS) is theoretically possible. This viewpoint is bolstered further by the selection of distinguishing characteristics. The emergence of distinctive wavelengths associated to the O-H chemical bond has been described before. Furthermore, the characteristic wavelengths in Figure 6 for the C-H chemical bond show in the absorption band 4484–4444 cm^−1^, which is related to the second overtone of C-H stretching. This shows that the model is responsive to pectin’s unique wavelengths, making it suitable for hardness modeling. Previous research has found large amounts of cellulose in passion fruit peel, which can be extracted and used to make nanofiber materials [23]. According to research, the area of about 7142 cm^−1^ may be ascribed to cellulose hydrogen bonds and O-H stretching, and chosen bands occur in this region in Figure 6 [24].

Table 4 presents the results of the GA-SVR and Grids-RF models constructed using feature bands selected in various ways. The Grids-RF model and the GA-SVR model developed using feature bands chosen using the CARS approach demonstrate constant predictive performance for the F1 parameter, with the Grids-RF model outperforming the GA-SVR model. The Grids-RF model, which was created using seven feature bands picked via MC preprocessing and CARS screening, performs extremely well, with a prediction set R^2^_P_ of 0.925 and RPD of 3.160, demonstrating its high ability to predict the F1 parameter.

In terms of F2 parameter prediction, the Grids-RF model with 14 feature bands determined using MA preprocessing and UVE algorithm screening has the strongest predictive performance, with R^2^_P_ and RPD values of 0.877 and 2.290, respectively. Although there is no notable increase over the Grids-RF model created via MA preprocessing with the complete 6001 wavelengths (R^2^_P_ of 0.908, RPD of 2.379), the smaller number of selected feature bands leads in a simpler model structure and higher detection efficiency.

In general, models built after screening using the SPA method perform badly in terms of prediction performance. This may be owing to the fact that the feature bands chosen by SPA are concentrated in a single absorption band, resulting in the extraction of critical information being incomplete.

### 3.5. Discussion of Final Models

Table 5 displays the ideal findings of the spectra after preprocessing and band selection for the GA-SVR and Grids-RF models, which were utilized for non-destructive detection of outer epicarp hardness (F1) and inner epicarp hardness (F2) of passion fruit. The findings reveal that whether for F1 or F2 prediction, the predictive ability of the existing Grids-RF models is greater. Specifically, the Grids-RF model developed with seven feature bands chosen using MC preprocessing and CARS screening (MC-CARS-Grids-RF) displays the greatest predictive performance for F1, with a prediction set R^2^_P_ of 0.925 and RPD of 3.160. The Grids-RF model created with MA preprocessing (MA-Grids-RF) demonstrates the greatest prediction performance for F2, with an R^2^_P_ of 0.908 and an RPD of 2.379.

Figure 7 displays the connection between the observed and predicted values of exterior F1 and F2 for the two best models indicated in Table 5.

## 4. Conclusions

This research utilizes Grids-RF and GA-SVR models, conducting comparisons across various preprocessing methods. Among these methods, the Grids-RF model, after MC preprocessing, performs optimally in predicting the F1 parameter of passion fruit, where the prediction set’s R^2^_P_ and RPD values achieved 0.898 and 2.867, respectively. The Grids-RF model, post-MA preprocessing, excels in F2 parameter performance, with R^2^_P_ and RPD values in the prediction set reaching 0.908 and 2.379, respectively. The performances of models based on CARS, SPA, and UVE band selection methods were concurrently compared. Results reveal that grids-RF models and GA-SVR models, established through the CARS method in conjunction with seven characteristic wavelengths, have predictive performance on the F1 parameter, with the Grids-RF model demonstrating superior predictability over the GA-SVR. For the F2 parameter, the MA-UVE-Grids-RF model of 14 characteristic wavelengths exhibited the best predictive efficacy, with R^2^_P_ and RPD values in the prediction set reaching 0.877 and 2.290, respectively.

Based on the band selection, bands within the absorption bands of 8403, 6896, 5154, 7142 cm^−1^, and 4484–4444 cm^−1^ were selected. These bands correspond to various movements of the O-H and C-H chemical bonds, unveiling the correlation between the foundation of near-infrared hardness detection and the change in hardness of passion fruit during storage. Ultimately, the most ideal predictive models were established for F1 and F2 parameters—the MC-CARS-Grids-RF model and the MA-Grids-RF model—which each achieved R^2^_P_ and RPD values of 0.925, 3.160 and 0.908, 2.379 in their respective prediction sets.

These findings underscore the capacity of near-infrared spectroscopy to predict the hardness of the pericarp of passion fruit, laying a foundation for quality identification. The significance of this study manifests in its substantiation of the accuracy and reliability of the use of diffuse reflectance to identify the interior and exterior hardness of passion fruit pericarp. Future research may consider the influence of variety differences in passion fruit and environmental conditions on hardness changes as well as the potential for near-infrared monitoring of pericarp hardness. However, regarding the limitations of our study, it is worth noting that while we used a variety of modelling methods to validate the data, our study results are based on a specific and relatively limited dataset. Potential regional differences and inherent differences between passion fruit varieties could have an underlying impact on the modelling results. Therefore, future research can include a broader range of passion fruit varieties and consider more environmental variables.

## Figures and Tables

**Figure 1 foods-13-00783-f001:**
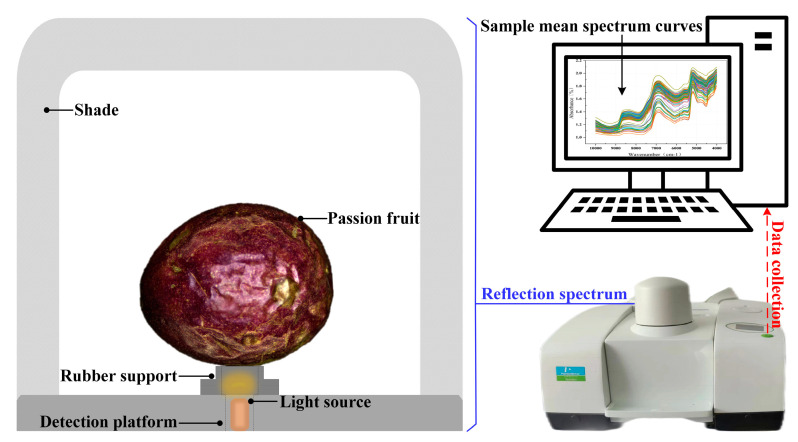
Diffuse reflection spectral acquisition system, including spectral inspection platform, instrument, and average spectral curves for all samples.

**Figure 2 foods-13-00783-f002:**
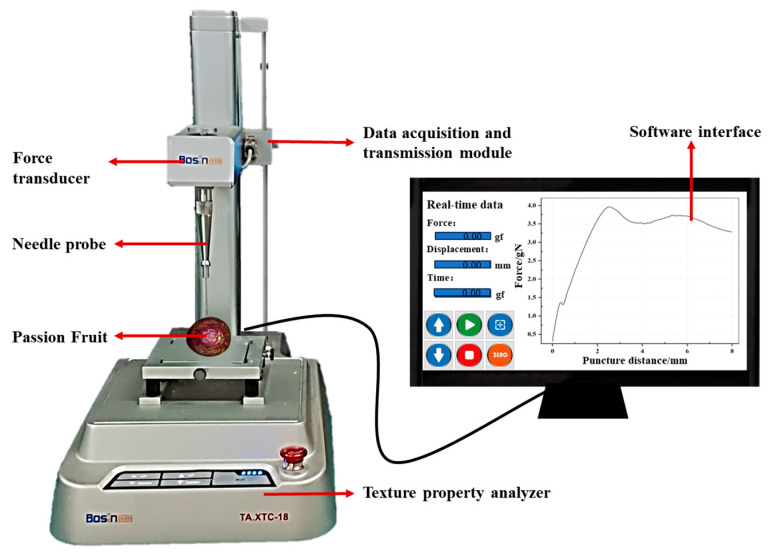
Schematic diagram of the texture analyzer.

**Figure 3 foods-13-00783-f003:**
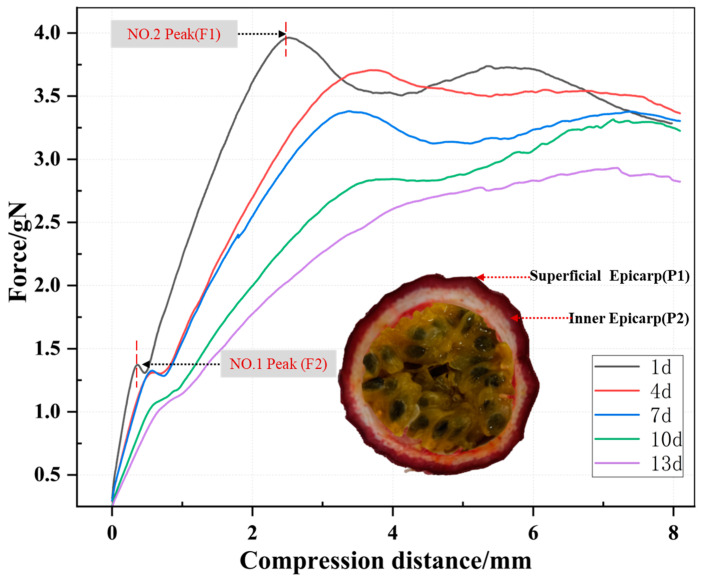
Variations in puncture force across different storage times. Each curve is obtained by averaging the characteristics of 20 samples from each batch and represents a force variation schematic curve with puncture displacement for that storage time.

**Figure 4 foods-13-00783-f004:**
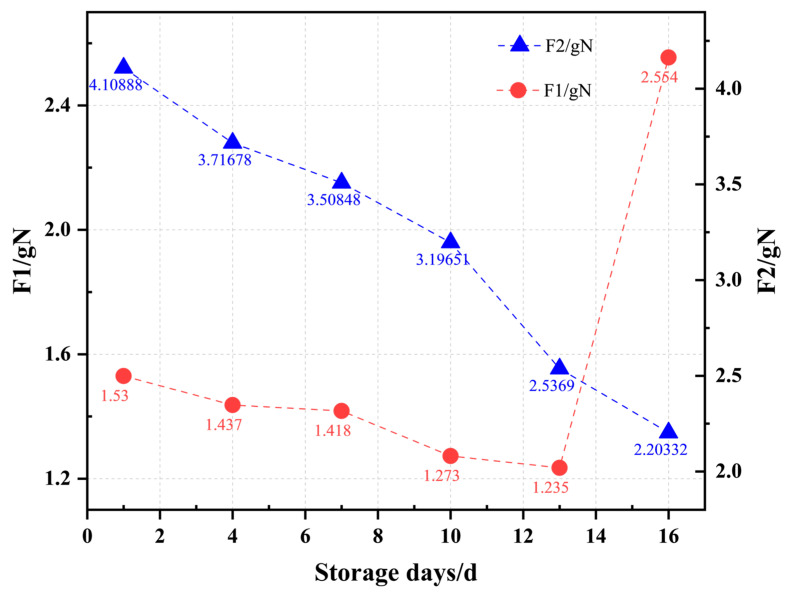
F1 and F2 under different storage durations. Each point is obtained by averaging the characteristics of 20 samples from each batch.

**Figure 5 foods-13-00783-f005:**
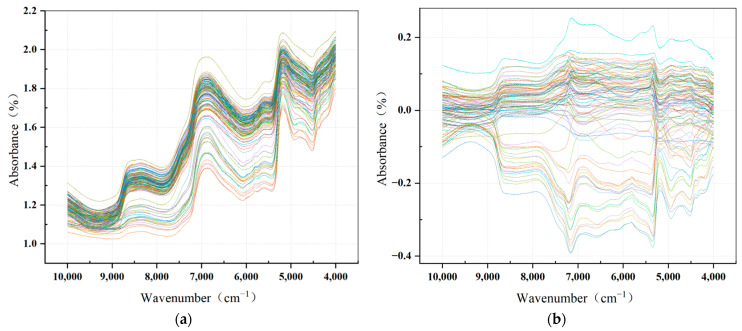
(**a**) Original Vis–NIR spectra curves of passion fruit samples; (**b**) Vis–NIR spectra after mean center pretreatment.

**Figure 6 foods-13-00783-f006:**
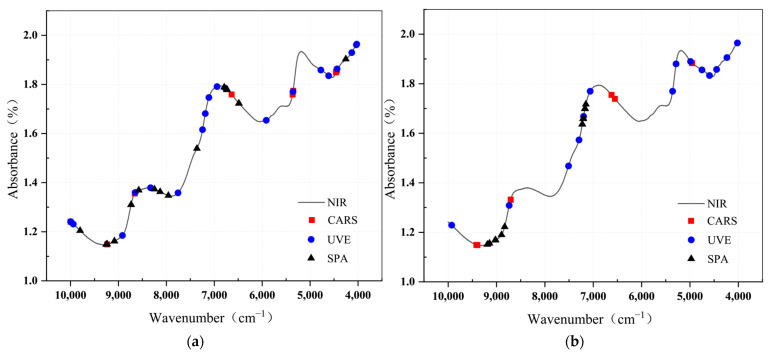
(**a**) Variables selected by CARS, UVE, and SPA of F1; (**b**) variables selected by CARS, UVE, and SPA F2.

**Figure 7 foods-13-00783-f007:**
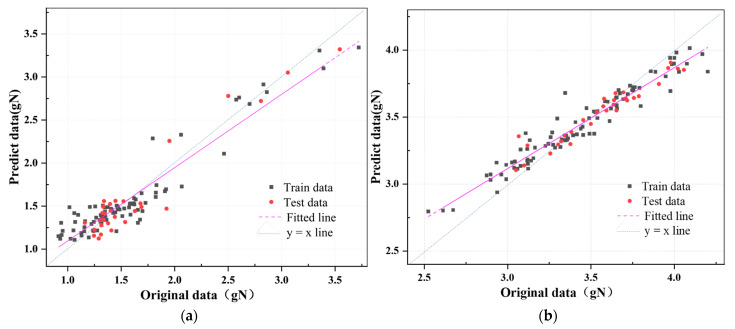
Best model for correlation between predicted values and original values of passion fruit puncture hardness. (**a**) MA-CARS-RF-Grids (5 folds) model of F1; (**b**) MA-Grids-RF (5 folds) model of F2.

**Table 1 foods-13-00783-t001:** Descriptive statistical results for F1 and F2 of passion fruit.

Variable	Minimum/gN	Maximum/gN	Mean/gN	Standard Deviation/gN	Number of Valid Cases/gN
F1	0.881	3.721	1.583	0.574	120
F2	2.524	4.201	3.464	0.356	120

**Table 2 foods-13-00783-t002:** Characteristics statistics of passion fruits extract samples during storage using SPXY algorithm to divide the sample set.

Variable	Calibration Set				Prediction Set			
	Minimum/gN	Maximum/gN	Mean/gN	Quantity	Minimum/gN	Maximum/gN	Mean/gN	Quantity
F1	0.881	3.721	1.548	90	1.064	3.544	1.646	30
F2	2.524	4.201	3.432	90	3.050	4.058	3.557	30

**Table 3 foods-13-00783-t003:** Different preprocessing data of F1 and F2 results for the Grids-RF (5 folds) and GA-SVR.

Variable	Model	Preprocessing	Calibration Set		Validation Set		RPD
			R^2^_C_	RMSE_C_/gN	R^2^_P_	RMSE_P_/gN	
F1	Grids-RF	RAW	0.869	0.208	0.874	0.211	2.706
	(5 folds)	MC	0.883	0.198	0.898	0.19	2.867
		MA	0.887	0.203	0.878	0.216	2.637
	GA-SVR	RAW	0.831	0.085	0.841	0.084	2.141
		MC	0.830	0.085	0.845	0.085	2.122
		MA	0.887	0.069	0.821	0.091	1.84
F2	Grids-RF	RAW	0.899	0.142	0.903	0.098	2.355
	(5 folds)	MC	0.797	0.167	0.743	0.142	1.576
		MA	0.912	0.127	0.908	0.101	2.379
	GA-SVR	RAW	0.817	0.1	0.834	0.079	2.003
		MC	0.747	0.118	0.777	0.079	2
		MA	0.815	0.1	0.865	0.068	2.079

**Table 4 foods-13-00783-t004:** Results of different models for predicting F1 and F2 in passion fruit samples.

Variable	Model	Method	Calibration Set		Validation Set		RPD
			R^2^_C_	RMSE_C_/gN	R^2^_P_	RMSE_P_/gN	
F1	Grids-RF (5folds)	MC-CARS	0.882	0.207	0.925	0.166	3.160
		MC-UVE	0.888	0.202	0.903	0.184	2.781
		MC-SPA	0.862	0.218	0.858	0.222	2.420
	GA-SVR	MC-CARS	0.897	0.143	0.901	0.075	2.773
		MC-UVE	0.890	0.144	0.818	0.122	2.012
		MC--SPA	0.909	0.133	0.744	0.158	1.411
F2	Grids-RF (5folds)	MA-CARS	0.882	0.147	0.868	0.110	2.209
		MA-UVE	0.895	0.143	0.877	0.109	2.290
		MA-SPA	0.833	0.168	0.746	0.155	1.208
	GA-SVR	MA-CARS	0.764	0.111	0.793	0.078	1.784
		MA-UVE	0.728	0.119	0.748	0.087	1.576
		MA-SPA	0.738	0.105	0.717	0.102	1.479

**Table 5 foods-13-00783-t005:** Results of different models for predicting F1 and F2 in passion fruit samples.

Dependent Variable	Model	Calibration Set		Validation Set		RPD
		R^2^_C_	RMSE_C_/gN	R^2^_P_	RMSE_P_/gN	
F1	MC-CARS-Grids-RF (5folds)	0.882	0.207	0.925	0.166	3.160
F2	MA-Grids-RF (5folds)	0.912	0.127	0.908	0.101	2.379

## Data Availability

The related data and methods are presented in this paper. Additional inquiries should be addressed to the corresponding author. The data are not publicly available due to the data are part of an ongoing study.

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
