# Peer review of "Nondestructive Determination of Epicarp Hardness of Passion Fruit Using Near-Infrared Spectroscopy during Storage"

_foods, 2024, doi:10.3390/foods13050783_

Round 1

Reviewer 1 Report

Comments and Suggestions for Authors

The present manuscript comprises a chemometric approach based on NIR spectroscopic data to evaluate the freshness of passion fruit by determining specific parameters, such as the hardness of the pericarp of passion fruit. The authors treated the data with different statistical techniques which support the initial idea of the study. In fact, the multivariate statistics models have been well explained. The figures are of good quality and the tables show all the statistical parameters. 

The manuscript design falls within the aims and scope of Foods MDPI journal. In addition, limited studies are available regarding passion fruit quality determination by using NIR and chemometrics. I support the study but the authors must revise it according to my attached suggestions within the pdf. Finally, the limitations and strengths of the study could be better highlighted through the manuscript.

Given my overall score of this paper, I suggest a minor revision prior to further consideration.

Comments on the Quality of English Language

The English language must be improved. The authors need to revise the grammar used.

Reviewer 2 Report

Comments and Suggestions for Authors

The manuscript is interesting, the description of the statistical methods used is sometimes too detailed.

I suggest some additions and modifications.

Reviewer 3 Report

Comments and Suggestions for Authors

The authors describe an interesting method to determine hardness of passion fruit.

the following revisions might be considered:

title: the flow of the title is not correct. Determination of epicarp hardness of passion fruit using …

line 12 and throughout: is superficial the correct English word?

line 22-23: rather state rmsep as result of prediction models

line 75: no firstnames in intent references?

line 87: incomplete sentence. Nir?

line 90: add speeches and variety. Were the samples from different farms? Add more details on sample variability

2.3: add validation data of reference method

line 126 and throughout: add space between number and unit

line 2.4: this reads like a textbook on multivatiate analysis. Avoid textbook knowledge. On the other hand, important information on the software packages and algorithms is missing. Please add.

table 1 and throughout: the units of f1 and f2 are mostly missing. Please add. Rmsep is also in real units.

line 385: incomplete?

line 388: explain, included instead of screened?

discussion: please improve discussion of limitations of method. Very restricted data set. Compare validation data to reference method.

Comments on the Quality of English Language

Mostly fine, minor editing required
